# Clinical Outcomes of Pediatric Chronic Intestinal Pseudo-Obstruction

**DOI:** 10.3390/jcm10112376

**Published:** 2021-05-28

**Authors:** Dayoung Ko, Hee-Beom Yang, Joong Youn, Hyun-Young Kim

**Affiliations:** 1Department of Pediatric Surgery, Seoul National University Children’s Hospital, Seoul 03080, Korea; kodayoung@gmail.com (D.K.); jkyoun@gmail.com (J.Y.); 2Department of Surgery, Seoul National University Bundang Hospital, Seongnam-si 13620, Korea; eeulere@naver.com; 3Department of Pediatric Surgery, Seoul National University College of Medicine, Seoul 03080, Korea

**Keywords:** chronic intestinal pseudo-obstruction, parenteral nutrition, pediatrics, myopathy, neuropathy

## Abstract

Chronic intestinal pseudo-obstruction (CIPO) is an extremely rare condition with symptoms of recurrent intestinal obstruction without any lesions. The outcomes of pediatric CIPO and predictors for the outcomes have not yet been well established. We analyzed the clinical outcomes and associated factors for the outcomes of pediatric CIPO. We retrospectively reviewed 66 primary CIPO patients diagnosed between January 1985 and December 2017. We evaluated parenteral nutrition (PN) factors such as PN duration, PN use over 6 months, home PN, and mortality as outcomes. We selected onset age, presence of urologic symptoms, pathologic type, and involvement extent as predictors. The early-onset CIPO was found in 63.6%, and 21.2% of the patients presenting with urologic symptoms. Of the 66 patients, 47 and 11 had neuropathy and myopathy, respectively. The generalized involvement type accounted for 83.3% of the cases. At the last follow-up, 24.2% of the patients required home PN management. The mean duration of PN was 11.8 ± 21.0 months. The overall mortality rate of primary CIPO was 18.2%. PN factors were predicted by the urologic symptoms and extent of involvement. However, mortality was predicted by pathologic type. The onset age was not significantly associated with the outcomes. CIPO with urologic symptoms and generalized CIPO had poor PN outcomes. Myopathy is suggested as a predictor of mortality in children with primary CIPO.

## 1. Introduction

Chronic intestinal pseudo-obstruction (CIPO) was first reported in 1958, and pediatric intestinal pseudo-obstruction (PIPO) was first reported in a case series of 11 children in 1977 [1]. When the patients show severe obstructive symptoms without any mechanical obstruction, we can suspect the possibility of CIPO. However, the diagnostic criteria for CIPOs varied in previous studies [2,3].

CIPO is divided into primary and secondary CIPOs. In the case of primary CIPO, neuropathy, myopathy or mesenchymopathy are shown as an abnormality in the enteric nervous system, not as a symptom of pseudo-obstruction caused by an underlying disease. However, pathogenesis of enteropathy is still not clearly established [4].

Generally, adult CIPOs tend to present in secondary forms, which are associated with systemic disease, and patients experience chronic abdominal pain. In contrast, PIPO present as a primary CIPO. Although many cases of PIPO develop as a sporadic form, several pathogenic mutations are reported [5]. The gene encoding the enteric smooth muscle contractile protein actin gamma 2 (ACTG2) are associated with a primary CIPO, visceral myopathy [6]. Mutation in the X-linked gene FLNA also associated with filaminopathy presented as a myopathic CIPO [7].

PIPO present with persistent vomiting and abdominal distension, which arises without any underlying cause [8]. The prognosis of CIPO is more aggressive in the pediatric population than in the adult population. In the pediatric population, growth failure is critical problem for the intestinal failure due to PIPO [5]. The rates of mortality and morbidity vary and remain unclear in PIPO [5,9,10]. The factors that are associated with mortality and morbidity in PIPO remain unclear.

In our study, we aimed to identify clinical manifestations, evaluate clinical outcomes, and analyze predictors of outcomes in PIPO.

## 2. Materials and Methods

### 2.1. Patients

According to the European Society for Paediatric Gastroenterology Hepatology and Nutrition (ESPGHAN)-Led Expert Group paper published in 2018, a pediatric primary CIPO was diagnosed when two or more of the following signs or symptoms were observed: (1) objective measure of small intestinal neuromuscular involvement, (2) recurrent and/or persistent bowel dilatation, (3) genetic and/or metabolic abnormality, and (4) inability to maintain adequate nutrition and/or growth upon oral feeding [5]. Owing to the different characteristics of PIPO, there is no clear unification of the diagnostic process. Recently, efforts have been made to unify diagnostic standards. The ESPGHAN society reported the diagnostic criteria for PIPO and recommended a step-by-step diagnostic approach, wherein obstructive symptoms caused by true obstruction and secondary causes among patients with abdominal distension could be excluded, and if two of the four diagnostic criteria were met, PIPO could be confirmed [5]. According to previous data, the “chronic” criterion is based on symptoms that persist for up to 2 months immediately after birth and thereafter for 6 months; other studies have reported that symptoms are based on a 6-month duration regardless of age [2,5,11].

Our hospital’s policy was to perform surgery when abdominal distension worsened 2 months before, and the patient had no choice but to undergo decompressive operation. The biopsy obtained at surgery confirmed the presence of ganglion cells and smooth muscle abnormalities. When the biopsy results were consistent with PIPO and showed persistent symptoms, PIPO was diagnosed and aggressive treatment was performed. Based on the diagnostic criteria of PIPO reported by the ESPGHAN-Led Expert Group, the data of 82 patients who visited Seoul National University Children’s Hospital and were suspected to have intestinal pseudo-obstruction from 1978 to 2017 were reviewed. We excluded 12 patients with mechanical obstruction. Four patients were excluded owing to secondary causes. In total, 66 patients with primary PIPOs were included in this study for analysis (Figure 1).

Out of the 82 patients who were suspected of having intestinal obstruction, 16 were excluded from this study because of mechanical obstruction and secondary causes. Finally, 66 primary PIPO patients were included for analysis.

This study was approved by the Institutional Review Board (IRB) of Seoul National University Hospital (IRB 1807-009-955).

### 2.2. Patients’ Characteristics

We retrospectively reviewed patients’ general characteristics including age, symptoms, pathology, extent of involvement, genetic mutation, operation, and clinical outcomes based on medical records. Clinical outcomes were evaluated using mortality and parenteral nutrition (PN) factors, which included PN duration, PN use over 6 months, and need for home PN.

### 2.3. Diagnostic Examinations

All specimens were examined by a dedicated pathologist and reviewed by another pathologist. Hematoxylin and eosin staining and immunohistochemistry were performed on full-thickness biopsy specimens. The pathologic type was categorized as neuropathy, myopathy, or undetermined. Neuropathy PIPO included hypoganglionosis and intestinal neuronal dysplasia type B. Myopathy PIPO was diagnosed when the specimen showed abnormality in the muscle layer, vacuolization of the muscle layer, additional muscle layer, and muscle degeneration with fibrosis (Figure 2).

In this study, 18 patients underwent whole exome sequencing analysis for identifying genetic mutations. Genetic testing included 13 genes known to be related to CIPO from previous studies: ACTA2, ACTG2, CLMP, FLNA, L1CAM, LMOD1, MYH11, MYLK, POLG, RAD21, SGOL1, SOX10, and TYMP.

### 2.4. Patient Groups

To analyze predictors for outcomes, known characteristics of PIPO including onset age group, urologic symptoms, pathology, and involvement extent were used in this study. The early-onset group was defined as a group of patients diagnosed with PIPO before the age of 1 month. Localized type involvement was defined when only one organ was invaded. The median follow-up period was 35 months.

### 2.5. Statistical Analyses

Continuous data were analyzed using the *t*-test. Categorical data were analyzed using the chi-squared test or Fisher exact test, as appropriate. Survival was evaluated using the Kaplan–Meier method and log-rank test. All statistical analyses were performed using software R version 3.4.0 (R Core Team, 2015). All tests were two-sided, and *p* values < 0.05 were considered statistically significant.

## 3. Results

The ratio of boys to girls was similar in our study. Symptoms developed around the age of 1 year (mean, 14.4 months). The number of early-onset patients with symptoms before the age of 1 month was 42 (63.6%). The most common initial symptom was abdominal distension (75.4%), and 21.2% of the patients presented urologic symptoms, including megacystis and vesicoureteral reflux at diagnosis (Table 1).

Forty-seven and 11 patients had neuropathy and myopathy, respectively. Of the 66 patients, 83.3% had generalized PIPOs, and 16.7% had localized PIPOs. The average number of operations per patient was 3.6. Most patients underwent enterostomy with intestinal biopsy (71.2%) (Table 2).

Among the 18 patients who underwent genetic analysis, four showed a mutation in the ACTG-2 gene, and one showed a mutation in SOX10 (Table 3).

The mean duration of PN use was 11.8 months, and 24.2% of the pediatric patients still required home PN at the last follow-up. The mortality rate was 18.2%, and the causes of death were sepsis, malnutrition, and multiorgan failure (Figure 3).

The mortality rate was significantly higher in patients with myopathic PIPOs than in patients with neuropathic PIPOs. However, symptom onset age, involvement type, and urologic symptoms were not associated with mortality in PIPOs. Regarding nutritional outcome, the total duration of PN was significantly longer in the generalized PIPO and in patients with urologic symptoms. The proportion of patients requiring PN over 6 months in the generalized group was 34.5%, which was significantly higher than that in patients with localized PIPO. Patients who presented with urologic symptoms also showed a higher proportion of PN usage over 6 months than patients without urologic symptoms. The proportion of patients requiring home PN had shown similar results, which was associated with urologic symptoms. In contrast, the pathologic type and age of onset had no association with nutritional outcomes (Table 4).

## 4. Discussion

Rudolph et al. have reported that PIPO could be diagnosed if intestinal obstruction symptoms and intestinal distension in plain abdominal radiographs persist without true obstruction or secondary causes [12]. PIPO is a heterogeneous condition with different causes, symptoms, and signs. For example, PIPO can be different for urologic symptoms and pathologic type [13]. Prior studies have reported that PIPO can combine urologic symptoms including megacystis and neurogenic bladder at birth [5]. Faure et al. have reported that, even in cases with normal biopsy results, megacystis could occur with neuropathy and myopathy [14]. Urological involvement rates of 36–100% are reported [15]. Our results showed 21% of urologic symptoms, which is consistent with the results of previous studies. Pathologic findings including neuropathy, myopathy, or non-specific findings could be observed in PIPO patients. According to Thapar et al., in PIPO, the neuropathy ratio was up to 70%; our study showed similar results at 71.2% [5].

Recent advancements in nutritional support and Intestinal Rehabilitation Programs (IRPs) improved the outcomes of intestinal failure including CIPO by lowering CIPO mortality rates. However, among the patients with intestinal failure, those with CIPO, which is a representative motility disorder, showed poorer outcomes than those with short bowel syndrome [16]. The survival rate of patients with short bowel syndrome is reported to be >95% since the IRPs, but the survival of patients with intestinal failure owing to CIPO is still reported to be approximately 85% [17,18,19]. In a previous study, CIPO outcomes were evaluated based on improvement after drug treatment, nutritional outcome, and death [19]. In our study, we evaluated mortality and nutritional outcomes. The mortality rate of CIPO was reported as 10–25% [18,19]. The cause of death was often owing to long-term PN complications including central catheter-associated sepsis and intestinal failure associated with liver disease [5]. The overall mortality rate of primary pediatric CIPO was 18.2%, which is in line with those of previous studies. All mortality cases occurred before 2011. At our institute, we performed a home PN program for patients with intestinal failure since 2008.

Despite our patients showing better survival, approximately 24% of them needed long-term PN and home PN management. Mousa et al. have reported that the rate of home PN in children diagnosed with CIPO was 60–80% [9]. In another study, one third of patients were dependent on home PN [19], which is higher than the number in the present study; this could be because it has not been long since our institute started home PN management. Among the 23 newly diagnosed CIPO patients from 2010, the rate of administering home PN was approximately 40%, compared to the previous 0%. We observed that 31.8% of patients relied on PN for ≥6 months. In a recent study, PN-dependent children showed low social quality of life. Particularly, patients receiving PN in the long term have negative emotions and limited sports activities. Caregivers also experience more depression, economic stress, and social isolation than those with children without CIPO [20]. Therefore, it is important to manage these patients comprehensively for improving mortality and nutritional outcomes.

Heneyke et al. have reported malrotation, short bowel, urinary involvement, and myopathy as poor prognostic factors for CIPO [10]. Another retrospective study of 105 pediatric patients has reported CIPO onset at birth, acute onset, megacystis, and operation as poor prognostic factors for PN dependence [14]. In the cases of early onset, especially before the age of 1 year, surgery is often performed for discriminating other obstructive causes; however, the outcomes are reported to be poor. Fell et al. have reported that only four out of 14 infants with CIPO recovered enteral autonomy, and five patients died, resulting in a mortality rate of 35.7% [21]. In contrast, there was no significant difference according to the onset age in current study. There was rare report regarding the onset time as a prognostic factor in recent 10 years. The previous results were also reported in the 1990s, and it is interpreted that the results such as mortality have improved as medical support including PN has been improved.

Urologic involvement is associated with diffuse hollow viscus organ involvement, suggesting the possibility of generalized disease rather than localized disease; it is reported as a predictor of poor outcome [14]. We identified that all 14 patients with urologic involvement had generalized type CIPO, and their nutritional outcome tended to be poor than patients without urologic symptoms. However, the mortality of patients with and without urologic symptoms was not significantly different, which might be because many patients use PN and can prevent sepsis due to bacterial translocation; however, the number of deaths due to the disease is low. Additionally, there was no difference in the nutritional outcome and mortality according to age at CIPO onset.

Pathologic type has been identified as an important prognostic factor for CIPO in many studies. In particular, the myopathy type was reported as a poor prognostic [10,22,23]. Additionally, patients with hypoganglionosis in pathologic specimens showed better survival rates than those without hypoganglionosis [24]. In our study, we analyzed the outcomes according to pathologic type: patients with myopathy had a significantly higher mortality rate (54.6%) than those with neuropathy (8.5%). This result was consistent with those of previous studies.

Kim et al. have reported good and fair outcomes in three localized CIPO cases, but four expired cases were reported in 19 generalized CIPOs when classified according to the involvement area [22]. In this study, according to the involvement type, mortality showed similar results with a previous study, which showed a difference in outcome according to the involvement area: the generalized type of 21.8% and localized type of 0%. However, it was not significant. Regarding nutritional outcomes for the localized type, there was no case in which PN was required for >6 months, and there was no case in which home PN was performed. Therefore, it was confirmed that the nutritional outcome was very poor in the generalized CIPO.

However, our study has some limitations. Although our results contain relatively large number of CIPO children, we collected data retrospectively. Future work should, therefore, include prospective study to evaluate risk factor.

## 5. Conclusions

In conclusion, we found that the diagnosis and proper management of pediatric primary CIPO are difficult to determine. However, appropriate management with a multidisciplinary approach and nutritional support could improve the mortality rates in CIPO. CIPO with myopathy is suggested to have poor mortality outcomes, and CIPO with urologic symptoms and generalized CIPO is suggested to have poor PN outcomes. It might be helpful in determining the treatment plan of PIPO patients based on the analysis of prognostic factors from this study.

## Figures and Tables

**Figure 1 jcm-10-02376-f001:**
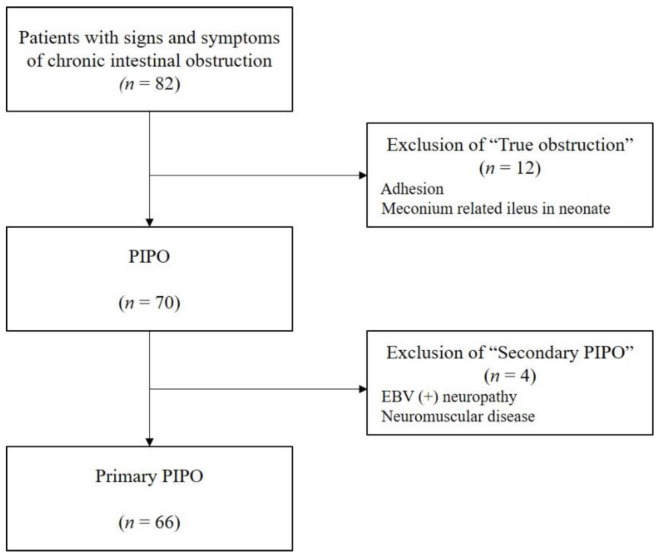
Diagnostic flow.

**Figure 2 jcm-10-02376-f002:**
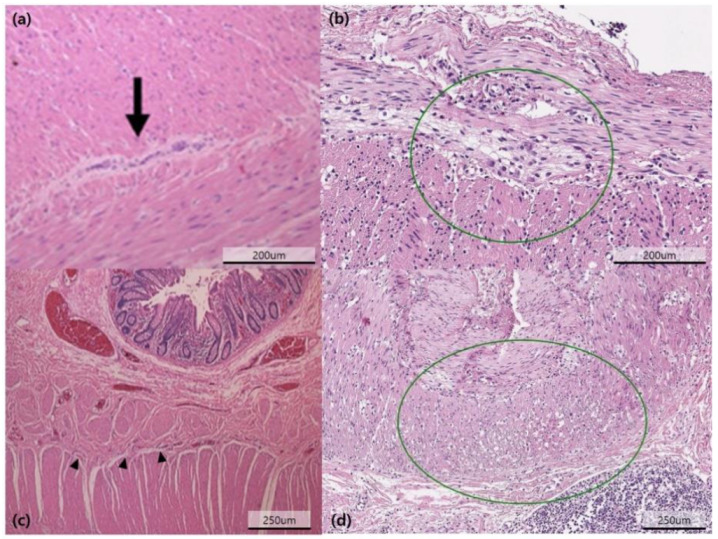
Pathologic specimens in chronic intestinal pseudo-obstruction patients were shown. Hematoxylin and eosin staining shows a hypoganglionosis. Arrow indicated the ganglion in the myenteric plexus (**a**); other slide demonstrates an immature ganglion cell (black triangle) (**b**); additional muscle layer was identified (circle) (**c**); muscle cells in the inner circular muscle layer show vacuolization (circle) (**d**).

**Figure 3 jcm-10-02376-f003:**
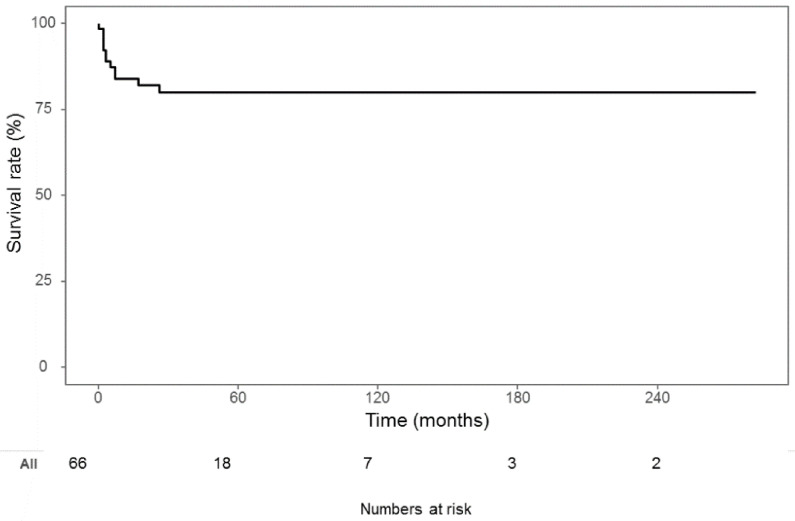
Overall survival was shown in this figure. Overall survival rate was 81.8%, and the median follow-up period was 35 months.

**Table 1 jcm-10-02376-t001:** General characteristics of pediatric chronic intestinal pseudo-obstruction patients.

	*n* (%)
Sex: male	34 (51.5%)
Birth weight (kg)	3.4 ± 0.5
Gestational age (day)	254.5 ± 28.5
Onset age (month)	14.4 ± 33.1
Early (≤1 month)	42 (63.6%)
Late (>1 month)	24 (36.4%)
Gastrointestinal symptom	
Abdominal distension	49 (75.4%)
Vomiting	29 (44.6%)
Constipation	19 (29.2%)
Feeding difficulty	7 (10.8%)
Diarrhea	4 (6.2%)
Abdominal pain	4 (6.2%)
Urologic symptom	14 (21.2%)
Megacystis	8 (12.1%)
Hydronephrosis	3 (4.5%)
Vesicoureteral reflux	2 (3.0%)
Neurogenic bladder	1 (1.5%)

**Table 2 jcm-10-02376-t002:** Disease characteristics in chronic intestinal pseudo-obstruction patients.

	*n* (%)
Pathology	
Neuropathy	47 (71.2%)
Hypoganglionosis	21 (31.8%)
IND-B ^§^	9 (13.6%)
Others	17 (25.8%)
Myopathy	11 (16.7%)
Neuropathy, myopathy	3 (4.6%)
Undetermined	5 (7.6%)
Extent of involvement	
Generalized	55 (83.3%)
Localized	11 (16.7%)
Stomach	3 (4.6%)
Small bowel	3 (4.6%)
Colon	5 (7.6%)
Genetic mutation	
ACTG-2 mutation	4 of 18
SOX10 mutation	1 of 18
CLMP, FLNA, MYH-11, RAD21, SGOL1	0 of 18
Number of operations	3.6 ± 2.2
Name of operation	
Bowel resection	
Gastrectomy	4
Colectomy	15
No bowel resection	
Full-thickness intestinal biopsy	2
Full-thickness intestinal biopsy with enterostomy	45
Outcome	
PN * duration (month)	11.8 ± 21.0
PN ≥ 6 months	21 (31.8%)
Home PN	16 (24.2%)
Mortality	12 (18.2%)

^§^ IND-B: intestinal neuronal dysplasia type B; * PN: parenteral nutrition.

**Table 3 jcm-10-02376-t003:** Genetic mutation.

Gene	Mutation
ACTG-2 mutation	c.533G > A, p.Arg178His, Heterozygote c.188G > A, p.Arg63Gln, Heterozygote c.188G > A, p.Arg63Gln, Heterozygote c.769C > T, p.Arg257Cys, Heterozygote
SOX10 mutation	c.1164T > A, p.Tyhr388, Heterozygote

**Table 4 jcm-10-02376-t004:** Comparison of clinical outcomes according to predictors.

	Onset Age			Urologic Symptom			Pathology					Involvement Extent		
	Early (*n* = 42)	Late (*n* = 24)	*p*	Yes (*n* = 14)	No (*n* = 52)	*p*	Neuropathy (*n* = 47)	Myopathy (*n* = 11)	Undetermined (*n* = 5)	N + M * (*n* = 3)	*p*	Generalized (*n* = 55)	Localized (*n* = 11)	*p*
PN duration	10 (23.8%)	9 (37.5%)	0.369	8.0 (2.0–48.0)	1.0 (0.0–6.0)	0.011	1.5 (0.0–19.0)	4.0 (1.0–34.0)	0.5 (0.0–2.0)	1.0 (0.5–40.5)	0.509	2.0 (0.0–15.5)	0 (0–0)	0.001
PN > 6 months	11 (26.2%)	5 (20.8%)	0.849	7 (50.0%)	12 (29.3%)	0.020	0 (0.0%)	4 (40.0%)	14 (36.8%)	1 (33.3%)	0.580	19 (34.5%)	0 (0.0%)	0.026
Home PN	7 (16.7%)	5 (20.8%)	0.928	9 (64.3%)	7 (13.5%)	<0.001	12 (25.5%)	2 (18.2%)	1 (20.0%)	1 (33.3%)	1.000	16 (29.1%)	0 (0.0%)	0.053
Mortality	3.4 ± 2.3	3.9 ± 2.1	0.394	1 (7.1%)	11 (21.2%)	0.436	4 (8.5%)	6 (54.5%)	2 (40.0%)	0 (0.0%)	0.002	12 (21.8%)	0 (0.0%)	0.199

* N + M: neuropathy and myopathy.

## Data Availability

Not applicable

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
