# Peer review of "Clinical Outcomes of Pediatric Chronic Intestinal Pseudo-Obstruction"

_jcm, 2021, doi:10.3390/jcm10112376_

Round 1

Reviewer 1 Report

Pediatric chronic intestinal pseudo-obstruction (CIPO) is the most life- threatening intestinal motility disorder and comprises a heterogeneous group of conditions that affect the function of intestinal neuro-musculature components. The paper by Ko and colleagues deals with the retrospective identification of factors that could affect outcome in pediatric CIPO patients. Using retrospective cohort, they found that CIPO with myopathy is associated with a high mortality rate to a poor mortality rate, and that CIPO with urologic symptoms and generalized CIPO is suggested to have poor Parenteral Nutrition outcome.

This data is of interest to the community but rewritting and the addition of additional data are necessary for the publication of this article

I have some comments and critics listed below.

1/ It is recommended a careful reading of the manuscript given that some sentences look not correct (mainly in the Abstract and Results section). Concerning the Results section, a detail of identified mutations in the CIPO patients could be included in a new table.

2/ Introduction need to be increase and more commented on the myopathy and neuropathy forms in pediatric CIPO; need to add more citations on the mutation identified in the CIPO patients

ACTG2 Holla et al, Endoscopy, 2014

FLNA Jenkins et al, Human Mutation 2018

MYH11 Araujo Moreno et al, Am J Med Genet A 2016

LMOD1 Halim et al, 2017, Proc Natl Acad Sci U S A. 2017 

MYLK Halim et al, Am J Human Genet 2017,

RAD21 Bonora et al Gastroenterology 2015

SGOL1 Chetaille et al Nat Genet 2014 

SOX10 Pingault V,  et al Ann Neurol. 2000 

Minors

1/ Figure 1: add scale bars, add high magnification

2/ Table 2: replace SGIL1 by SGOL1

Author Response

I have some comments and critics listed below.

  • It is recommended a careful reading of the manuscript given that some sentences look not correct (mainly in the Abstract and Results section). Concerning the Results section, a detail of identified mutations in the CIPO patients could be included in a new table.

Answer:

Additional description regarding the genetic mutation was added in the results section. (page 5, Table 3)

  • Introduction need to be increase and more commented on the myopathy and neuropathy forms in pediatric CIPO; need to add more citations on the mutation identified in the CIPO patients

Answer:

We intensified the introduction section as you mentioned. (page 2)

ACTG2 Holla et al, Endoscopy, 2014

FLNA Jenkins et al, Human Mutation 2018

MYH11 Araujo Moreno et al, Am J Med Genet A 2016

LMOD1 Halim et al, 2017, Proc Natl Acad Sci U S A. 2017 

MYLK Halim et al, Am J Human Genet 2017,

RAD21 Bonora et al Gastroenterology 2015

SGOL1 Chetaille et al Nat Genet 2014 

SOX10 Pingault V,  et al Ann Neurol. 2000 

Minors

1) Figure 1: add scale bars, add high magnification

 Answer: We revised the figure as you commented. However, we could not add a high magnification version.

2) Table 2: replace SGIL1 by SGOL1

 Answer: We corrected the word from SGIL1 to SGOL1. (page 5)

Reviewer 2 Report

Here are my comments regarding the manuscript "Which factors could affect the outcome of chronic intestinal pseudo-obstruction in children?" by Dayoung et al.

The article is easy to understand and provides descriptive data on a rare disease, pediatric intestinal pseudo-obstrucion. The data is descriptive and retrospective weakening the results.

My main question to the authors is: what is new in this study?

My comments:

1. The authors refer to the consensus statement by ESPGHAN and in this consensus it is suggested to use the term "PIPO" in pediatric CIPO patients. I suggest that the authors change this.

2. The title is a sentence with a question mark. I would change this to a statement such as "Outcomes in pediatric intestinal pseudo obstruction" or something in similar fashion.

3. The introduction is very short and should be more elaborate. For example, parenteral nutrition and it's problems should be at least be mentioned.

4. In introduction the sentence "The prognosis of CIPO is more aggressive in the pediatric population than in the adult population." is unclear. Do you mean mortality is higher or progresses faster? Please specify.

5. In subheading "Diagnostic examination" the authors use the term "pathologic type". This is a confusing term, please be more specific and use term "histolopathological diagnosis" or something similar. Use the same term then in Table 2 and further in text.

6. In the first chapter of discussion, please state your main findings clearly. What is new in this study?? Now you just state that this is similar as previous studies. Why should this be published????

7. The discussion is overall very long and should be reduced. 

Eg.

  • Why chapters 2 and 4 (rows 152-170) are separate? Are these relevant in discussion? This seems to me like explanation of methods and should be in the methods section?
  • Chapter 5, rows 171-179, why is this relevant in this study? This study did not investigate the usefulness of surgery??
  • Chapter 6, rows 180-182, why is this data relevant?
  • Chapters 7 and 8 should be combined.
  • Chapter 10, rows 211-219, please state the findings of this study and not only other studies. 

8. In conclusion, please transfer the discussion about limitations to discussion.

9. State in conclusions what is new in this study.

Author Response

The article is easy to understand and provides descriptive data on a rare disease, pediatric intestinal pseudo-obstrucion. The data is descriptive and retrospective weakening the results.

My main question to the authors is: what is new in this study?

My comments:

  1. The authors refer to the consensus statement by ESPGHAN and in this consensus it is suggested to use the term "PIPO" in pediatric CIPO patients. I suggest that the authors change this.

Answer:

We corrected the term as you commented

  1. The title is a sentence with a question mark. I would change this to a statement such as "Outcomes in pediatric intestinal pseudo obstruction" or something in similar fashion.

Answer:

We corrected the term as you commented

  1. The introduction is very short and should be more elaborate. For example, parenteral nutrition and it's problems should be at least be mentioned.

Answer:

We corrected the introduction including the complication related to the PIPO.  

  1. In introduction the sentence "The prognosis of CIPO is more aggressive in the pediatric population than in the adult population." is unclear. Do you mean mortality is higher or progresses faster? Please specify.

Answer:

we corrected the term as you commented

  1. In subheading "Diagnostic examination" the authors use the term "pathologic type". This is a confusing term, please be more specific and use term "histolopathological diagnosis" or something similar. Use the same term then in Table 2 and further in text.
  2. In the first chapter of discussion, please state your main findings clearly. What is new in this study?? Now you just state that this is similar as previous studies. Why should this be published????
  3. The discussion is overall very long and should be reduced. 

Eg.

  • Why chapters 2 and 4 (rows 152-170) are separate? Are these relevant in discussion? This seems to me like explanation of methods and should be in the methods section?
  • Chapter 5, rows 171-179, why is this relevant in this study? This study did not investigate the usefulness of surgery??
  • Chapter 6, rows 180-182, why is this data relevant?
  • Chapters 7 and 8 should be combined.
  • Chapter 10, rows 211-219, please state the findings of this study and not only other studies. 
  1. In conclusion, please transfer the discussion about limitations to discussion.
  2. State in conclusions what is new in this study.

Round 2

Reviewer 2 Report

Authors have made significant improvements to the manuscript, I have no further comments.

I support the acceptance of the manuscript by Dayoung et al.